# Zero-Resource Multilingual Model Transfer: Learning What to Share

## Abstract

Modern natural language processing and understanding applications have enjoyed a great boost utilizing neural networks models. However, this is not the case for most languages especially low-resource ones with insufficient annotated training data. Cross-lingual transfer learning methods improve the performance on a low-resource target language by leveraging labeled data from other (source) languages, typically with the help of cross-lingual resources such as parallel corpora. In this work, we propose a *zero-resource* multilingual transfer learning model[1] that can utilize training data in multiple source languages, while not requiring *target language training data* nor *cross-lingual supervision*. Unlike most existing methods that only rely on language-invariant features for cross-lingual transfer, our approach utilizes both language-invariant and language-specific features in a coherent way. Our model leverages adversarial networks to learn language-invariant features and mixture-of-experts models to dynamically exploit the relation between the target language and each individual source language. This enables our model to learn effectively what to share between various languages in the multilingual setup. It results in significant performance gains over prior art, as shown in an extensive set of experiments over multiple text classification and sequence tagging tasks including a large-scale real-world industry dataset.

## 1 Introduction

The recent deep learning revolution enables a wide variety of NLP models to achieve impressive performance, thanks in part to large-scale annotated datasets. However, such an advantage is not available to most of the world languages since only a handful of them have the labeled data necessary for training deep neural nets. As it is prohibitive to obtain training data for all languages of interest, *cross-lingual transfer learning* (CLTL) comes to the rescue to enable learning models for a *target language* using annotated data from other languages (*source languages*) (Yarowsky et al., 2001).

In this paper, we study the more challenging *unsupervised* CLTL setting, where *no* target language labeled data is used for training.[2] In this setting, most previous work relies on cross-lingual resources in one form or another in order to transfer models across languages, such as bilingual lexica (Mihalcea et al., 2007), parallel corpora (Yarowsky et al., 2001), or machine translation systems (Wan, 2009). In contrast, this work proposes a *zero-resource* CLTL framework that relies on *no* cross-lingual resources whatsoever. In addition, we focus on a multi-source CLTL, also known as multilingual transfer learning (MLTL) scenario that can leverage labeled data in multiple source languages simultaneously to improve the performance on the low-resource target language.

Distinct from other transfer learning tasks such as domain adaptation, one unique difficulty faced by cross-lingual transfer learning is the disparate input space problem represented by disjoint sets of vocabulary which cripples the use of traditional feature representations. Therefore, the CLTL problem typically consists of two parts: *cross-lingual language representation*, and *model transfer*. Fortunately, recent research in unsupervised learning of cross-lingual word embeddings (Lample et al., 2018) provides a viable solution for the language representation problem without the need for parallel corpora. We hence focus on the model transfer problem in this work.

---

[1]The code will be available at http://[url redacted for anonymity].
[2]In contrast, supervised CLTL assumes the availability of annotations in the target language.

The most straightforward method for cross-lingual model transfer is through weight sharing, namely directly applying the model trained on the source language to the target language. However, as shown in previous work (Chen et al., 2016), the feature distributions of different languages extracted by the same neural net are still dissimilar, and weight sharing is not sufficient for learning language-invariant features that generalize well across languages. Existing work, therefore, typically relies on *language-adversarial training* (Chen et al., 2016; Kim et al., 2017) to extract features that are invariant with respect to the shift in language, using only unlabeled texts from each language.

Nonetheless, in the MLTL setting, where multiple source languages exist, language-adversarial training will only use for model transfer the features that are common among all source languages and the target, which may be too restrictive in many cases. For example, when transferring from English, Spanish and Chinese to German, language-adversarial training will retain only features that are invariant across all four languages, which can be too sparse to be informative. On the other hand, the fact that German is more similar to English than to Chinese is neglected and the transferred model is unable to utilize features that are shared only between English and German. To address these shortcomings, we propose a new model that not only exploits language-invariant features, but also allows the target language to dynamically and selectively leverage language-specific features through a probabilistic attention-style mixture of experts mechanism (see Section 3). This allows our model to learn what to share between various languages.

For multiple CLTL tasks ranging from text classification to sequence tagging including a real-world large-scale industry dataset, our model beats all baseline models trained, like ours, without cross-lingual supervision. More strikingly, it could in many cases match or outperform state-of-the-art models that have access to strong cross-lingual supervision (e.g. commercial machine translation systems or millions of parallel sentences).

## 2 RELATED WORK

The diversity of human languages is a critical challenge for natural language processing. In order to alleviate the need for obtaining annotated data for each task in each language, cross-lingual transfer learning (CLTL) has long been studied (Yarowsky et al., 2001; Bel et al., 2003, *inter alia*).

One CLTL direction is the *supervised* setting, where training data is available in the target language, and the goal is to further boost the performance by resorting to labeled data in additional languages. In the presence of target language training data, recent work in deep learning is able to perform CLTL without relying on additional cross-lingual resources (Kim et al., 2017; Yang et al., 2017).

On the other hand, an arguably more challenging setting is the *unsupervised* setting, where no target language training data is available. Traditionally, research focuses on resource-based methods, where general-purpose cross-lingual resources such as MT systems or parallel corpora are utilized to replace task-specific annotated data (Wan, 2009; Prettenhofer & Stein, 2010). Zhang et al. (2016) could use as few as ten word translation pairs for CLTL, but is restricted to the part-of-speech tagging task. With the advent of deep learning, especially adversarial neural networks (Goodfellow et al., 2014; Ganin et al., 2016), progress has been made towards model-based CLTL methods. Chen et al. (2016) propose language-adversarial training that does not directly depend on parallel corpora, but instead only requires a set of bilingual word embeddings (BWEs). However, the BWEs used in their work were still trained using parallel corpus.

Another important direction for CLTL is to learn cross-lingual word representations (Klementiev et al., 2012; Zou et al., 2013; Mikolov et al., 2013). Recently, there have been several notable work for learning fully unsupervised cross-lingual word embeddings, both for the bilingual (Zhang et al., 2017; Lample et al., 2018; Artetxe et al., 2018) and multilingual case (Chen & Cardie, 2018b). These efforts pave the road for performing CLTL without cross-lingual resources.

## 3 MODEL

One commonly adopted paradigm for neural CLTL models is the *shared-private* model (Bousmalis et al., 2016; Kim et al., 2017), where the features are divided into two parts: *shared* (language-invariant) features and *private* (language-specific) features. As mentioned before, the shared features are enforced to be language-invariant via language-adversarial training, by attempt-

ing to fool a language discriminator. Furthermore, Chen & Cardie (2018a) propose a generalized shared-private model for the multi-source setting, where a *multinomial adversarial network* (MAN) is adopted to extract common features shared by all source languages as well as the target.

On the other hand, the private features are learned by separate feature extractors, one for each source language, capturing the remaining features outside the shared ones. During training, the labeled samples from a certain source language go through the corresponding private feature extractor for that particular language. At test time, there is no private feature extractor for the target language; only the shared features are used for cross-lingual transfer.

As mentioned in Section 1, using only the shared features for model transfer imposes an overly strong constraint and many useful features may be wiped out by adversarial training if they are shared only between the target language and a subset of source languages. Therefore, we propose to use a mixture-of-experts (MoE) model (Shazeer et al., 2017; Gu et al., 2018) to learn the private features. The high-level idea is to have a set of language expert networks, one per source language, each responsible for learning language-specific features for that source language during training. However, instead of hard-switching between the experts, each sample uses a convex combination of all experts, dictated by an *expert gate*. Thus, at test time, the trained expert gate can decide what combination to use for the unseen target language based on its similarity to the source languages.

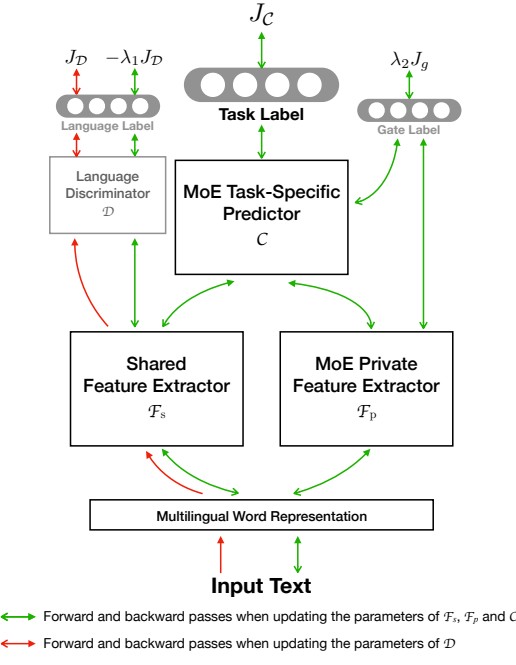

Figure 1: An overview of the MAN-MoE model.

experts, dictated by an *expert gate*. Thus, at test time, the trained expert gate can decide what combination to use for the unseen target language based on its similarity to the source languages.

Figure 1 shows an overview of our full MAN-MoE model for multilingual model transfer. The boxes illustrate various components of the MAN-MoE model (§3.1), while the arrows depict the training flow. (More training details can be found in Appendix A.)

## 3.1 Model Architecture

Figure 1 portrays an abstract view of the MAN-MoE model with four major components: the Multilingual Word Representation, the Shared Feature Extractor $\mathcal{F}_s$ (together with the Language Discriminator $\mathcal{D}$), the MoE Private Feature Extractor $\mathcal{F}_p$, and finally the MoE Predictor $\mathcal{C}$. Based on the actual task (e.g. sequence tagging, text classification, sequence to sequence, etc.) different architectures may be adopted, as explained below.

**Multilingual Word Representation** embeds words from all languages into a single semantic space so that words with similar meanings are close to each other regardless of language. In this work, we mainly rely on the MUSE embeddings (Lample et al., 2018) that are fully unsupervised. We map all other languages into English to obtain a multilingual embedding space. However, in certain experiments, MUSE yields 0 accuracy on one or more language pairs (Søgaard et al., 2018), in which case the VecMap embeddings (Artetxe et al., 2017) are used. It uses *identical strings* as supervision, which does not require parallel corpus or human annotations to train.

In addition, for tasks where morphological features are important, one can add character-level word embeddings (Dos Santos & Zadrozny, 2014) that captures sub-word information. When character embeddings are used, we add a single CharCNN that is shared across all languages, and the final word representation is the concatenation of the word embedding and the char-level embedding. The CharCNN can then be trained end to end with the rest of the model.

**Shared Feature Extractor** $\mathcal{F}_s$ is a multinomial adversarial network (Chen & Cardie, 2018a), which is an adversarial pair of a feature extractor (e.g. LSTM or CNN) and a Language Discriminator $\mathcal{D}$. $\mathcal{D}$ is a text classifier (Kim, 2014) that takes the shared features (extracted by $\mathcal{F}_s$) of an input sequence

and predicts which language it comes from. On the other hand, $\mathcal{F}_s$ strives to fool $\mathcal{D}$ so that it cannot identify the language of a sample. The hypothesis is that if $\mathcal{D}$ cannot recognize the language of the input, the shared features then do not contain language information and are hence language-invariant. Note that $\mathcal{D}$ is trained only using unlabeled corpus, and can therefore be trained on all languages including the target language with no labeled data.

**MoE Private Feature Extractor** $\mathcal{F}_p$ is a key difference of our model from previous work, which is shown in Figure 2. The figure shows the Mixture-of-Experts (Shazeer et al., 2017) model with three source languages, English, German and Spanish. $\mathcal{F}_p$ has a shared BiL-STM at the bottom that extracts contextualized word representations for each token $w$ in the input sentence. The LSTM hidden representation $\boldsymbol{h}_w$ is then fed into the MoE module, where each source language has a separate expert network (a MLP). In addition, the *expert gate* $\mathcal{G}$ is a linear transformation that takes $\boldsymbol{h}_w$ as input and outputs a softmax score $\alpha_i$ for each expert. The final private feature vector is a mixture of all expert outputs, dictated by the expert gate weights $\alpha$. During training, the gate loss $J_g$ is used to encourage samples from a certain source language to use the correct expert (see Appendix A for more details), and each expert is hence

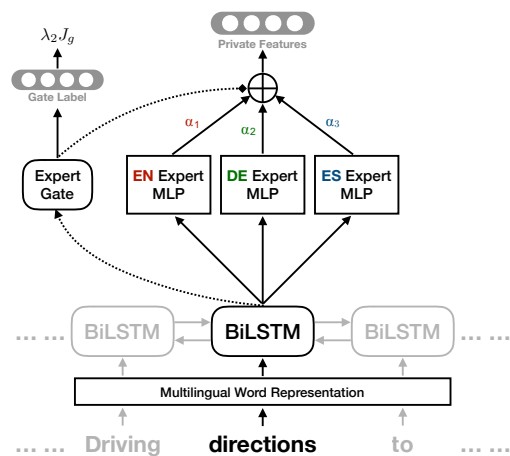

Figure 2: The Private Feature Extractor $\mathcal{F}_p$.

learning language-specific features for that language. At test time, the trained expert gate will examine the hidden representation of a token, and produces the optimal expert weights. Therefore, $\mathcal{F}_p$ is able to dynamically determine what knowledge to use at a token level, serving as a much more flexible and powerful feature extractor for those features that are not shared across all languages.

**MoE Task-Specific Predictor** $\mathcal{C}$ is the final module that make predictions for the end task, and may take different forms depending on the task. For instance, Figure 3 shows the MoE predictor for sequence tagging, where one output label is predicted for each input token. It is straightforward to adapt $\mathcal{C}$ to work for other tasks. For example, for text classification, a pooling layer such as dot-product attention (Luong et al., 2015) can be added at the bottom to fuse token-level features into a single sentence feature vector.

$\mathcal{C}$ first concatenates the shared and private features to form a single feature vector for each token. It then has another Mixture-of-Experts module that outputs a softmax probability over all labels for each token. The idea is that it may be favorable to put different weights between the language-invariant and language-specific features for different target languages. Again consider the example of English, German, Spanish and Chinese. When transferring to Chinese from the other three, the source languages are similar to each other while all being rather

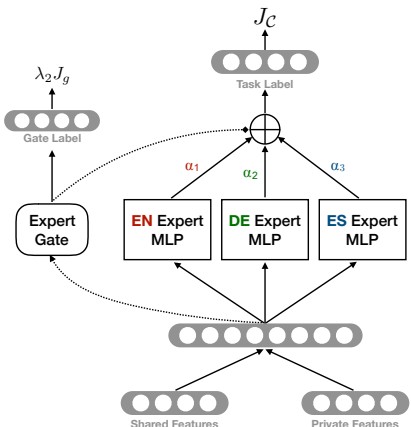

Figure 3: The MoE Predictor $\mathcal{C}$ for Sequence Tagging.

distant from Chinese. Therefore, the adversarially learned shared features might be more important in this case. On the other hand, when transferring to German, which is much more similar to English than to Chinese, we might want to pay more attention to the MoE private features. Therefore, we adopt a MoE module in $\mathcal{C}$, which provides more flexibility than using a single MLP.[3]

---

[3] We also experimented with an attention mechanism between the shared and private features in $\mathcal{C}$, or adding a gating mechanism to modulate each feature channel, but adding another MoE in $\mathcal{C}$ gave the best results.

| Domain | English | | | German | | | Spanish | | | Chinese | | | #Slot |
|---|---|---|---|---|---|---|---|---|---|---|---|---|---|
| | #Train | #Dev | #Test | #Train | #Dev | #Test | #Train | #Dev | #Test | #Train | #Dev | #Test | |
| Navigation | 311045 | 23480 | 36625 | 13356 | 1599 | 2014 | 13862 | 1497 | 1986 | 7472 | 1114 | 1173 | 8 |
| Calendar | 64010 | 5946 | 8260 | 8261 | 1084 | 1366 | 6706 | 926 | 1081 | 2056 | 309 | 390 | 4 |
| Files | 30339 | 2058 | 5355 | 3005 | 451 | 480 | 6082 | 843 | 970 | 1289 | 256 | 215 | 5 |

| Domain | Examples |
|---|---|
| Navigation | $[Driving]_{transportation\_type}$ directions to $[Walmart]_{place\_name}$ in $[New\ York]_{location}$. |
| Calendar | Add $[school\ meeting]_{title}$ to my calendar on $[Monday]_{start\_date}$ at $[noon]_{start\_time}$. |
| Files | Search for $[notes]_{data\_type}$ with $[grocery\ list]_{keyword}$. |

Table 1: Statistics for the personal virtual assistant slot filling dataset with examples.

## 4 EXPERIMENTS

In this section, we present an extensive set of experiments across three datasets. The first experiment is on a large-scale real-world multilingual slot filling (sequence tagging) dataset, where the data is used in a commercial personal virtual assistant. In addition, we conduct experiments on two public academic datasets, namely the CoNLL 2002/2003 Multilingual Named Entity Recognition (sequence tagging) dataset (Sang, 2002; Sang & Meulder, 2003), and the Multilingual Amazon Reviews (text classification) dataset (Prettenhofer & Stein, 2010).

### 4.1 CROSS-LINGUAL SLOT FILLING FOR VIRTUAL ASSISTANTS

As shown in Table 1, we collect data for four languages: English, German, Spanish, and Chinese, over three domains: Navigation, Calendar, and Files. Each domain has a set of pre-determined slots (the slots are the same across languages), and the user utterances in each language and domain are annotated by crowd workers with the correct slots (see the examples in Table 1). We employ the standard BIO tagging scheme to formulate the slot filling problem as a sequence tagging task.

For each domain and language, the data is divided into a training, a validation, and a test set, with the corresponding number of samples in each split shown in Table 1. We can see that there is a natural imbalance in the amount of available data for each language, which further motivates cross-lingual transfer learning. In our experiments, we treat each domain as a separate experiment, and consider each of German, Spanish and Chinese as the target language while the remaining three being source languages, which results in a total of 9 experiments.

#### 4.1.1 RESULTS

In Table 2, we report the performance of `MAN-MoE` compared to a number of baseline systems. All systems adopt the same base architecture, which is a multi-layer BiLSTM sequence tagger (İrsoy & Cardie, 2014) with a token-level MLP on top (no CRFs used).

**MT baselines** employ machine translation (MT) for cross-lingual transfer. In particular, the *train-on-trans(lation)* method translates the entire English training set into each target language which are in turn used to train a supervised system on the target language. On the other hand, the *test-on-trans(lation)* method trains an English sequence tagger, and utilizes MT to translate the test set of each target language into English in order to make predictions. In this work, we adopt the Microsoft Translator[4], a strong commercial MT system. Note that for a MT system to work for sequence tagging tasks, *word alignment* information must be available, in order to project word-level annotations across languages. This rules out many MT systems such as Google Translate since they do not provide word alignment information through their APIs.

**BWE baselines** rely on Bilingual Word Embeddings (BWEs) and weight sharing for CLTL. Namely, the sequence tagger trained on the source language(s) are directly applied to the target language, in hopes that the BWEs could bridge the language gap. This simple method has been shown to yield strong results in recent work (Upadhyay et al., 2018). The MUSE (Lample et al., 2018) BWEs are

---

[4]https://azure.microsoft.com/en-us/services/cognitive-services/translator-text-api/

| Domain | German | | | | Spanish | | | | Chinese | | | |
|---|---|---|---|---|---|---|---|---|---|---|---|---|
| | Navi. | Cal. | Files | *avg.* | Navi. | Cal. | Files | *avg.* | Navi. | Cal. | Files | *avg.* |
| *Methods with cross-lingual resources* | | | | | | | | | | | | |
| MT (train-on-trans.) | 59.95 | 63.53 | 38.68 | 54.05 | 64.37 | **59.93** | 67.55 | **63.95** | **60.56** | **66.49** | 61.01 | **62.69** |
| MT (test-on-trans.) | 54.49 | 51.74 | 55.87 | 54.03 | 52.13 | 58.10 | 55.00 | 55.08 | 54.23 | 22.71 | **64.01** | 46.98 |
| *Methods without cross-lingual resources* | | | | | | | | | | | | |
| BWE (1-to-1) | 57.53 | 58.28 | 35.73 | 50.51 | 62.54 | 44.44 | 57.56 | 54.85 | 17.62 | 22.48 | 21.32 | 20.47 |
| BWE (3-to-1) | 61.03 | 67.66 | 51.30 | 60.00 | 63.74 | 45.10 | 64.47 | 57.77 | 20.91 | 13.70 | 28.47 | 21.03 |
| MAN | 59.07 | 60.24 | 39.35 | 52.89 | 58.86 | 37.90 | 46.75 | 47.84 | 34.45 | 13.53 | 40.63 | 29.54 |
| MAN-MoE | **62.73** | **75.13** | 59.19 | **65.68** | **66.57** | 50.21 | **70.91** | 62.56 | 34.18 | 29.36 | 41.70 | 35.08 |

Table 2: F1 scores on the slot filling dataset. The highest performance is in bold, while the highest performance within method group (with vs. without cross-lingual supervision) is underlined.

| Domain | German | | | | Spanish | | | | Chinese | | | |
|---|---|---|---|---|---|---|---|---|---|---|---|---|
| | Navi. | Cal. | Files | *avg.* | Navi. | Cal. | Files | *avg.* | Navi. | Cal. | Files | *avg.* |
| MAN-MoE | 62.73 | 75.13 | **59.19** | **65.68** | **66.57** | 50.21 | **70.91** | **62.56** | 34.18 | **29.36** | 41.70 | **35.08** |
| - Tagger MoE | **63.42** | **76.68** | 55.68 | 65.26 | 65.50 | 47.51 | 69.67 | 60.89 | 27.71 | 21.75 | **41.77** | 30.41 |
| - Private MoE | 58.33 | 48.85 | 37.35 | 48.18 | 58.99 | 36.67 | 48.39 | 48.02 | **39.61** | 14.64 | 38.08 | 30.78 |
| - MoE | 59.07 | 60.24 | 39.35 | 52.89 | 58.86 | 37.90 | 46.75 | 47.84 | 34.45 | 13.53 | 40.63 | 29.54 |
| - MAN | 60.64 | 67.69 | 55.10 | 61.14 | 65.38 | 46.71 | 68.25 | 60.11 | 18.43 | 10.82 | 28.90 | 19.38 |

Table 3: Ablation results on the slot filling dataset.

used by all systems in this experiment. *1-to-1* indicates that we are only transferring from English, while *3-to-1* means the training data from all other three languages are leveraged.[5]

The final baseline is the MAN model (Chen & Cardie, 2018a), presented before our MAN-MoE approach. As shown in Table 2, MAN-MoE substantially outperforms all baseline systems that do not employ cross-lingual supervision on almost all domains and languages. Another interesting observation is that MAN performs strongly on Chinese while being much worse on German and Spanish compared to the BWE baseline. This corroborates our hypothesis that MAN only leverages features that are invariant across *all* languages for CLTL, and it learns such features better than weight sharing. Therefore, when transferring to German or Spanish, which is similar to a subset of source languages, the performance of MAN degrades significantly. On the other hand, when Chinese serves as the target language, where all source languages are rather distant from it, MAN has its merit in extracting language-invariant features that could generalize to Chinese. With MAN-MoE, however, this trade-off between close and distant language pairs is well addressed by the combination of MAN and MoE. By utilizing both language-invariant and language-specific features for transfer, MAN-MoE outperforms all cross-lingually unsupervised baselines on all languages.

Furthermore, even when compared with the MT baseline, which has access to hundreds of millions of parallel sentences, MAN-MoE performs competitively on German and Spanish. It even significantly beats both MT systems on German as MT sometimes fails to provide word alignment for German. On Chinese, where the unsupervised BWEs are much less accurate (BWE baselines only achieve 20% F1), MAN-MoE is able to greatly improve over the BWE and MAN baselines and shows promising results for fully unsupervised CLTL even between distant language pairs.

### 4.1.2 FEATURE ABLATION

In this section, we take a closer look at the various modules of MAN-MoE and their impacts on performance. When the tagger ($\mathcal{C}$) MoE is removed, moderate decrease is observed on all languages. The performance degrades the most on Chinese, suggesting that using a single MLP in $\mathcal{C}$ is not ideal when the target language is not similar to the sources. When removing the private MoE, the MoE in $\mathcal{C}$ no longer makes much sense as $\mathcal{C}$ only has access to the shared features, and the performance is

---

[5]MAN and MAN-MoE results are always 3-to-1 in this paper.

even slightly worse than removing both MoEs. With both MoE modules removed, it reduces to the MAN model, and we saw a significant drop on German and Spanish. Finally, when removing MAN while keeping MoE, where the shared features are simply learned via weight-sharing, we see a slight drop on German and Spanish, but a rather great one on Chinese. The ablation results support our hypotheses and validate the merit of MAN-MoE.

## 4.2  CROSS-LINGUAL NAMED ENTITY RECOGNITION

In this section, we present experiments on the CoNLL 2002/2003 multilingual named entity recognition (NER) dataset (Sang, 2002; Sang & Meulder, 2003), with four languages: English, German, Spanish and Dutch. The task is also formulated as a sequence tagging problem, with four types of tags: PER, LOC, ORG, and MISC.

The results are summarized in Table 4. We observe that using only word embeddings does not yield satisfactory results, since the out-of-vocabulary problem is rather severe, and morphological features such as capitalization is crucial for NER. We hence add character-level word embeddings for this task (§3.1) to capture subword features and alleviate the OOV problem. For German, however, all nouns are capitalized, and the capitalization features

| Target Language | de | es | nl | avg |
|---|---|---|---|---|
| *Methods with cross-lingual resources* | | | | |
| Täckström et al. (2012) | 40.4 | 59.3 | 58.4 | 52.7 |
| Nothman et al. (2013) | 55.8 | 61.0 | 64.0 | 60.3 |
| Tsai et al. (2016) | 48.1 | 60.6 | 61.6 | 56.8 |
| Ni et al. (2017) | **58.5** | 65.1 | 65.4 | 63.0 |
| Mayhew et al. (2017) | 57.5 | 66.0 | 64.5 | 62.3 |
| *Methods without cross-lingual resources* | | | | |
| MAN-MoE | 55.1 | 59.5 | 61.8 | 58.8 |
| BWE+CharCNN (1-to-1) | 51.5 | 61.0 | 67.3 | 60.0 |
| BWE+CharCNN (3-to-1) | 55.8 | 70.4 | 69.8 | 65.3 |
| Xie et al. (2018)[*] | 56.9 | 71.0 | 71.3 | 66.4 |
| MAN-MoE+CharCNN | 56.7 | 71.0 | 70.9 | 66.2 |
| MAN-MoE+CharCNN+UMWE | 56.0 | **73.5** | **72.4** | **67.3** |

[*] Contemporaneous work

Table 4: F1 scores for the CoNLL NER dataset on German (de), Spanish (es) and Dutch (nl).

learned on the other three languages would lead to poor results. Therefore, for German only, we lowercase all characters in systems that adopt CharCNN.

Table 4 also shows state-of-the-art models in the literature. Note that most of these systems are specifically designed for the NER task, and exploit many task-specific resources, such as multilingual gazetteers, or the metadata in Freebase or Wikipedia (such as entity categories). Among these, Täckström et al. (2012) rely on parallel corpora to learn cross-lingual word clusters that serve as features. Nothman et al. (2013); Tsai et al. (2016) both leverage the structured and unstructured information in external knowledge bases such as Wikipedia to learn useful features for cross-lingual NER. Ni et al. (2017) employ noisy parallel corpora (aligned sentence pairs, but are not always translations) as well as bilingual dictionaries (5k words for each language pair) for model transfer. They further added external features such as entity types learned from Wikipedia for improved performance. Finally, Mayhew et al. (2017) propose a multi-source framework that utilizes large cross-lingual lexica. Despite not using any of these resources, general or task-specific, MAN-MoE nonetheless outperforms all these methods. The only exception is German, where task-specific resources may still be helpful, due to its high OOV rate and unique capitalization rules.

In a contemporaneous work by (Xie et al., 2018), they propose a cross-lingual NER model using Bi-LSTM-CRF that achieves similar performance compared to MAN-MoE+CharCNN. However, our architecture is not specialized to the NER task, and we did not add task-specific modules such as a CRF decoding layer, etc.

Last but not least, we replace the MUSE embeddings with the recently proposed unsupervised multilingual word embeddings (Chen & Cardie, 2018b), which further boosts the performance, achieving a new state-of-the-art performance.

## 4.3  CROSS-LINGUAL TEXT CLASSIFICATION ON AMAZON REVIEWS

Finally, we report results on a multilingual text classification dataset (Prettenhofer & Stein, 2010). The dataset is a binary classification dataset where each review is classified into positive or negative sentiment. It has four languages: English, German, French and Japanese.

| Domain | German | | | | French | | | | Japanese | | | |
|---|---|---|---|---|---|---|---|---|---|---|---|---|
| | books | dvd | music | *avg* | books | dvd | music | *avg* | books | dvd | music | *avg* |
| *Methods with general-purpose cross-lingual resources* | | | | | | | | | | | | |
| MT-BOW[1] | 79.68 | 77.92 | 77.22 | 78.27 | 80.76 | 78.83 | 75.78 | 78.46 | 70.22 | 71.30 | 72.02 | 71.18 |
| CL-SCL[1] | 79.50 | 76.92 | 77.79 | 78.07 | 78.49 | 78.80 | 77.92 | 78.40 | 73.09 | 71.07 | 75.11 | 73.09 |
| CR-RL[2] | 79.89 | 77.14 | 77.27 | 78.10 | 78.25 | 74.83 | 78.71 | 77.26 | 71.11 | 73.12 | 74.38 | 72.87 |
| Bi-PV[3] | 79.51 | 78.60 | **82.45** | 80.19 | **84.25** | 79.60 | 80.09 | 81.31 | 71.75 | 75.40 | 75.45 | 74.20 |
| UMM[4] | 81.65 | 81.27 | 81.32 | 81.41 | 80.27 | 80.27 | 79.41 | 79.98 | 71.23 | 72.55 | 75.38 | 73.05 |
| *Methods with task-specific cross-lingual resources* | | | | | | | | | | | | |
| CLDFA[5] | **83.95** | **83.14** | 79.02 | **82.04** | 83.37 | 82.56 | **83.31** | **83.08** | **77.36** | **80.52** | **76.46** | **78.11** |
| *Methods without cross-lingual resources* | | | | | | | | | | | | |
| BWE (1-to-1) | 76.00 | 76.30 | 73.50 | 75.27 | 77.80 | 78.60 | 78.10 | 78.17 | 55.93 | 57.55 | 54.35 | 55.94 |
| BWE (3-to-1) | 78.35 | 77.45 | 76.70 | 77.50 | 77.95 | 79.25 | 79.95 | 79.05 | 54.78 | 54.20 | 51.30 | 53.43 |
| MAN-MoE | 82.40 | 78.80 | 77.15 | 79.45 | 81.10 | **84.25** | 80.90 | 82.08 | 62.78 | 69.10 | 72.60 | 68.16 |

[1] Prettenhofer & Stein (2010)  [2] Xiao & Guo (2013)  [3] Pham et al. (2015)
[4] Xu & Wan (2017)  [5] Xu & Yang (2017)

Table 5: Results for the Multilingual Amazon Review Text Classification dataset. Numbers indicate binary classification accuracy. VecMap embeddings (Artetxe et al., 2017) are used for this experiment as MUSE training fails on Japanese (see §3.1).

As shown in Table 5, MT-BOW uses machine translation to translate the bag of words of a target sentence into the source language, while CL-SCL learns a cross-lingual feature space via structural correspondence learning (Prettenhofer & Stein, 2010). CR-RL (Xiao & Guo, 2013) learns bilingual word representations where part of the word vector is shared among languages. Bi-PV (Pham et al., 2015) extracts bilingual paragraph vector by sharing the representation between parallel documents. UMM (Xu & Wan, 2017) is a multilingual framework that could utilize parallel corpora between multiple language pairs, and pivot as needed when direct bitexts are not available for a specific source-target pair. Finally CLDFA (Xu & Yang, 2017) proposes cross-lingual distillation on parallel corpora for CLTL. Unlike other works listed, however, they adopt a task-specific parallel corpus (translated Amazon reviews) that are difficult to obtain in practice, making the numbers not directly comparable to others.

Among these methods, UMM is the only one that does not require direct parallel corpus between all source-target pairs. It could instead utilize pivot languages (e.g. English) to connect multiple languages. MAN-MoE, however, takes another giant leap forward to completely remove the necessity of parallel corpora while achieving similar results on German and French compared to UMM. On Japanese, the performance of MAN-MoE is again limited by the quality of BWEs. (BWE baselines are merely better than randomness.) Nevertheless, MAN-MoE remains highly effective and the performance is only a few points below most methods with cross-lingual supervision.

## 4.4 VISUALIZATION OF EXPERT GATE WEIGHTS

In Figure 4, we visualize the average expert gate weights for each of the three target languages in the Amazon dataset. For each sample, we first compute a sentence-level aggregation by averaging over the expert gate weights of all its tokens. These sentence-level expert gate weights are then further averaged across all samples in the validation set of all three domains (books, dvd, music), which forms a final language-level average expert gate weight for each target language.

The visualization further collaborates with our hypothesis that our model makes informed decisions when selecting what features to share to the target language. It can be seen that when transferring to German or French (from the remaining three), the Japanese expert is less utilized compared to the European languages. On the other hand, it is interesting that when transferring to Japanese, the French and English experts are used more than the German one, and the exact reason remains to be investigated. However, this phenomenon might be of less significance since the private features may not play a very important role when transferring to Japanese as the model is probably focusing more on the shared features, according to the ablation study in §4.1.2.

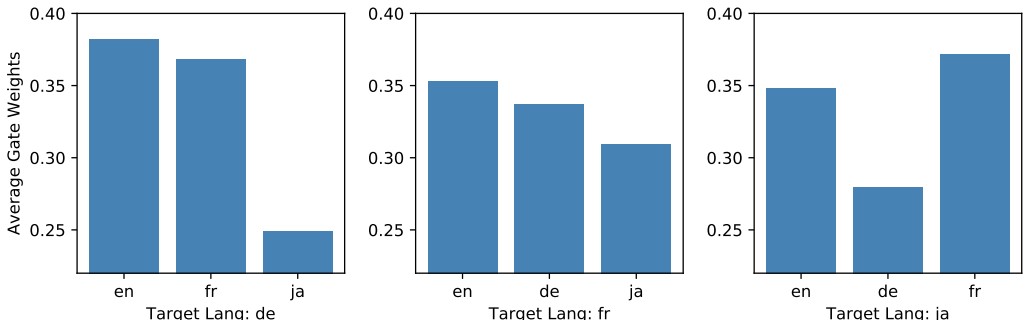

Figure 4: Average expert gate weights aggregated on a language level for the Amazon dataset.

## 5    CONCLUSION

In this paper, we propose a zero-resource multilingual model transfer approach that requires neither target language training data nor general-purpose cross-lingual resources. Our `MAN-MoE` method exploits both language-invariant (shared) features and language-specific (private) features for CLTL, which departs from previous models that could only make use of shared features. Following earlier work, the shared features are learned via language-adversarial training (Chen et al., 2016). However, the key difference is that we employ a Mixture-of-Experts (`MoE`) module to extract the private features, which is able to dynamically capture the relation between the target language and each source language on a token level. This is extremely helpful when the target language is similar to a subset of source languages, in which case traditional models that solely rely on shared features would perform poorly.

Our claim is supported by a wide range of experiments over multiple text classification and sequence tagging tasks, including a large-scale real-world industry dataset. `MAN-MoE` significantly outperforms all cross-lingually unsupervised baselines regardless of task or language. Furthermore, even considering methods with strong cross-lingual supervision (e.g. commercial machine translation systems or millions of parallel sentences), `MAN-MoE` is able to match or outperform these models on closer language pairs. On distant language pairs such as English-Chinese or English-Japanese, where the quality of cross-lingual word embeddings are unsatisfactory, `MAN-MoE` remains highly effective and substantially mitigates the performance gap introduced by cross-lingual supervision.

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

## APPENDIX A   MODEL TRAINING

Denote the set of all $N$ source languages as $\mathcal{S}$, where $|\mathcal{S}| = N$. Denote the target language as $\mathcal{T}$, and let $\Delta = \mathcal{S} \cup \mathcal{T}$ be the set of all languages. Denote the annotated corpus for a source language $l \in \mathcal{S}$ as $\mathbb{X}_l$, where $(x, y) \sim \mathbb{X}_l$ is a sample drawn from $\mathbb{X}_l$. In addition, unlabeled data is required for all languages to facilitate the MAN training. We hence denote as $\mathbb{U}_{l'}$ the unlabeled texts from a language $l' \in \Delta$.

---

**Algorithm 1** MAN−MoE Training

---

**Require:** labeled corpus $\mathbb{X}$; unlabeled corpus $\mathbb{U}$; Hyperpamameter $\lambda_1, \lambda_2 > 0$, $k \in \mathbb{N}$

1: **repeat**
2:      ▷ $\mathcal{D}$ iterations
3:      **for** $diter = 1$ to $k$ **do**
4:          $l_{\mathcal{D}} = 0$
5:          **for all** $l \in \Delta$ **do**                                                                  ▷ For all languages
6:              Sample a mini-batch $\boldsymbol{x} \sim \mathbb{U}_l$
7:              $\boldsymbol{f}_s = \mathcal{F}_s(\boldsymbol{x})$                                                       ▷ Shared features
8:              $l_{\mathcal{D}} \mathrel{+}= L_{\mathcal{D}}(\mathcal{D}(\boldsymbol{f}_s); l)$                                            ▷ $\mathcal{D}$ loss
9:          Update $\mathcal{D}$ parameters using $\nabla l_{\mathcal{D}}$
10:      ▷ Main iteration
11:      $loss = 0$
12:      **for all** $l \in \mathcal{S}$ **do**                                                          ▷ For all source languages
13:          Sample a mini-batch $(\boldsymbol{x}, \boldsymbol{y}) \sim \mathbb{X}_l$
14:          $\boldsymbol{f}_s = \mathcal{F}_s(\boldsymbol{x})$                                                       ▷ Shared features
15:          $\boldsymbol{f}_p, \boldsymbol{g}_1 = \mathcal{F}_p(\boldsymbol{x})$                                    ▷ Private features and gate outputs
16:          $\hat{\boldsymbol{y}}, \boldsymbol{g}_2 = \mathcal{C}(\boldsymbol{f}_s, \boldsymbol{f}_p)$
17:          $loss \mathrel{+}= L_{\mathcal{C}}(\hat{\boldsymbol{y}}; \boldsymbol{y}) + \lambda_2(L_g(\boldsymbol{g}_1; l) + L_g(\boldsymbol{g}_2; l))$                     ▷ $\mathcal{C}$ loss and gate loss
18:      **for all** $l \in \Delta$ **do**                                                                  ▷ For all languages
19:          Sample a mini-batch $\boldsymbol{x} \sim \mathbb{U}_l$
20:          $\boldsymbol{f}_s = \mathcal{F}_s(\boldsymbol{x})$                                                       ▷ Shared features
21:          $loss \mathrel{+}= -\lambda_1 \cdot L_{\mathcal{D}}(\mathcal{D}(\boldsymbol{f}_s); l)$                              ▷ Language loss to confuse $\mathcal{D}$
22:      Update $\mathcal{F}_s, \mathcal{F}_p, \mathcal{C}$ parameters using $\nabla loss$
23: **until** convergence

---

The overall training flow of variant components is illustrated in Figure 1, while the training algorithm is depicted in Algorithm 1. Similar to MAN, there are two separate optimizers to train MAN−MoE, one updating the parameters of $\mathcal{D}$ (red arrows), while the other updating the parameters of all other modules (green arrows). In Algorithm 1, $L_{\mathcal{C}}$, $L_{\mathcal{D}}$ and $L_g$ are the loss functions for the predictor $\mathcal{C}$, the language discriminator $\mathcal{D}$, and the expert gates in $\mathcal{F}_p$ and $\mathcal{C}$, respectively.

In practice, we adopt the NLL loss for $L_{\mathcal{C}}$ for text classification, and token-level NLL loss for sequence tagging:

$$L^{NLL}(\hat{y}; y) = -\log P(\hat{y} = y) \tag{1}$$

$$L^{T\text{-}NLL}(\hat{\boldsymbol{y}}; \boldsymbol{y}) = -\log P(\hat{\boldsymbol{y}} = \boldsymbol{y}) = -\sum_i \log P(\hat{y}_i = y_i) \tag{2}$$

where $y$ is a scalar class label, and $\boldsymbol{y}$ is a vector of token labels. $L_{\mathcal{C}}$ is hence interpreted as the negative log-likelihood of predicting the correct task label. Similarly, $\mathcal{D}$ adopts the NLL loss in (1) for predicting the correct language of a sample. Finally, the expert gates $\mathcal{G}$ use token-level NLL loss in (2), which translates to the negative log-likelihood of using the correct language expert for each token in a sample.

|             | $\lambda_1$ | $\lambda_2$ | $k$ |
|-------------|-------------|-------------|-----|
| Slot Filling | 0.01       | 1           | 5   |
| CoNLL NER   | 0.0001      | 0.01        | 1   |
| Amazon      | 0.002       | 0.1         | 1   |

Table 6: The hyperparameter choices for different experiments.

Therefore, the objectives that $\mathcal{C}$, $\mathcal{D}$ and $\mathcal{G}$ minimize are, respectively:

$$J_{\mathcal{C}} = \sum_{l \in \mathcal{S}} \mathop{\mathbb{E}}_{(x,y) \in \mathbb{X}_l} \left[ L_{\mathcal{C}}(\mathcal{C}(\mathcal{F}_s(x), \mathcal{F}_p(x)); y) \right] \tag{3}$$

$$J_{\mathcal{D}} = \sum_{l \in \Delta} \mathop{\mathbb{E}}_{x \in \mathbb{U}_l} \left[ L_{\mathcal{D}}(\mathcal{D}(\mathcal{F}_s(x)); l) \right] \tag{4}$$

$$J_{\mathcal{G}} = \sum_{l \in \mathcal{S}} \mathop{\mathbb{E}}_{x \in \mathbb{X}_l} \left[ \sum_{w \in x} L_{\mathcal{G}}(\mathcal{G}(h_w); l) \right] \tag{5}$$

where $\boldsymbol{h}_w$ in (5) is the BiLSTM hidden representation in $\mathcal{F}_p$ as shown in Figure 2. In addition, note that $\mathcal{D}$ is trained using unlabeled corpora over all languages ($\Delta$), while the training of $\mathcal{F}_p$ and $\mathcal{C}$ (and hence $\mathcal{G}$) only take place on source languages ($\mathcal{S}$). Finally, the overall objective function is:

$$J = J_{\mathcal{C}} - \lambda_1 J_{\mathcal{D}} + \lambda_2 (J_{\mathcal{G}}^{(1)} + J_{\mathcal{G}}^{(2)}) \tag{6}$$

where $J_{\mathcal{G}}^{(1)}$ and $J_{\mathcal{G}}^{(2)}$ are the two expert gates in $\mathcal{F}_p$ and $\mathcal{C}$, respectively.

## APPENDIX B    IMPLEMENTATION DETAILS

In all experiments, Adam (Kingma & Ba, 2015) is used for both optimizers (main optimizer and $\mathcal{D}$ optimizer), with learning rate $0.001$ and weight decay $10^{-8}$. Batch size is 64 for the slot filling experiment and 16 for the NER and Amazon Reviews experiments, which is selected mainly due to memory concerns. CharCNN increases the GPU memory usage and NER hence could only use a batch size of 16 to fit in 12GB of GPU memory. The Amazon experiment does not employ character embeddings but the documents are much longer, and thus also using a smaller batch size. All embeddings are fixed during training. Dropout (Srivastava et al., 2014) with $p = 0.5$ is applied in all components. Unless otherwise mentioned, ReLU is used as non-linear activation.

Bidirectional-LSTM is used in the feature extractors for all experiments. In particular, $\mathcal{F}_s$ is a two-layer BiLSTM of hidden size 128 (64 for each direction), and $\mathcal{F}_p$ is a two-layer BiLSTM of hidden size 128 stacked with a MoE module (see Figure 2). Each expert network in the MoE module of $\mathcal{F}_p$ is a two-layer MLP again of hidden size of 128. The final layer in the MLP has a $tanh$ activation instead of ReLU to match the LSTM-extracted shared features (with $tanh$ activations). The expert gate is a linear transformation (matrix) of size $128 \times N$, where $N$ is the number of source languages.

On the other hand, the architecture of the task specific predictor $\mathcal{C}$ depends on the task. For sequence tagging experiments, the structure of $\mathcal{C}$ is shown in Figure 3, where each expert in the MoE module is a token-level two-layer MLP with a softmax layer on top for making token label predictions. For text classification tasks, a dot-product attention mechanism (Luong et al., 2015) is added after the shared and private features are concatenated. It has a length 256 weight vector that attends to the feature vectors of each token and computes a softmax mixture that pools the token-level feature vectors into a single sentence-level feature vector. The rest of $\mathcal{C}$ remains the same for text classification.

For the language discriminator $\mathcal{D}$, a CNN text classifier (Kim, 2014) is adopted in all experiments. It takes as input the shared feature vectors of each token, and employs a CNN with max-pooling to pool them into a single fixed-length feature vector, which is then fed into a MLP for classifying the language of the input sequence. The number of kernels is 200 in the CNN, while the kernel sizes are 3, 4, and 5. The MLP has one hidden layer of size 128.

The MUSE and VecMap embeddings are trained with the monolingual $300d$ fastText Wikipedia embeddings (Bojanowski et al., 2017). When character-level word embeddings are used, a Char-CNN is added that takes randomly initialized character embeddings of each character in a word,

and passes them through a CNN with kernel number 200 and kernel sizes 3, 4, and 5. Finally, the character embeddings are max-pooled and fed into a single fully-connected layer to form a 128 dimensional character-level word embedding, which is concatenated with the pre-trained cross-lingual word embedding to form the final word representation of that word.

The remaining hyperparameters such as $\lambda_1$, $\lambda_2$ and $k$ (see Algorithm 1) are tuned for each individual experiment, as shown in Table 6.

