# OpenReview forum: "Zero-Resource Multilingual Model Transfer: Learning What to Share"
_ICLR.cc/2019/Conference_

### Official Review · AnonReviewer2 · 2018-10-27
**Nice idea but not very novel, extensive evaluation setup with positive resutls, unawareness of some important previos work**

**Rating:** 6
**Confidence:** 4

**Review:**

This paper describes a model for cross lingual transfer with no target language information. This is a well written paper that makes a number of contributions:

1. It provides an interesting discussion of transfer form multiple source languages into a target language. This is a timely problem and the paper points out that adversarial networks may be too limiting in this setup.

2. It provides a modeling approach that deals with the limitations of adversarial networks as mentioned in (1).

3. It demonstrates the value of the proposed approach through an extensive experimental setup.

At the same time, I see two major limitations to the paper:

1. While the proposed approach is valid, it is not very original, at least in my subjective eyes. The authors integrate a classifier that combines the private, language-specific features so that not only features that are shared between all the involved languages can be used in the classification process. While this is a reasonable idea that works well in practice, IMO it is quite straight forward and builds on ideas that have been recently been proposed in many other works.

2. The authors claim that: "To our best knowledge, this work is the first to propose
an unsupervised CLTL framework without depending on any cross-lingual resource"

This is, unfortunately, not true. I refer the authors to the paper:

Deep Pivot-Based Modeling for Cross-language Cross-domain Transfer with Minimal Guidance. Yftah Ziser and Roi Reichart. EMNLP 2018.

In their lazy setup, the EMNLP authors do exactly that. They address the more complicated cross-language, cross-domain setup, but their model can be easily employed within a single domain. Their experiments even use the multilingual sentiment dataset used in the current paper. The model in the EMNLP paper shows to outperform adversarial networks, so it can be competitive here as well.

---

> ### Author Response · Authors · 2018-11-13
> **Response to the reviewer's comments**
>
> Thank you for your comments and feedback.
>
> Regarding the limitations:
> 1: We believe the originality of our model lies in the fact that, unlike previous work, it is able to coherently utilize both the shared and the private features when transferring from multiple sources. In particular, our model dynamically determines what knowledge to share to the target language on a token-level basis. In addition, the usage of the mixture-of-experts model in a transfer learning setting is also novel to our knowledge.
>
> 2: In the Ziser and Reichart paper, the authors used the multilingual embeddings from Smith et al. (https://github.com/Babylonpartners/fastText_multilingual), which according the following github description, is not unsupervised: "Of the 89 languages provided by Facebook, 78 are supported by the Google Translate API. We first obtained the 10,000 most common words in the English fastText vocabulary, and then use the API to translate these words into the 78 languages available. We split this vocabulary in two, assigning the first 5000 words to the training dictionary, and the second 5000 to the test dictionary."
> In addition, we would like to ask the reviewer to kindly consider the fact that EMNLP 2018 proceedings were not available until the end of October, which was one month after the ICLR submission deadline.
> Nevertheless, we agree with the reviewer that it is not necessary to emphasize our method being the first zero-resource CLTL method, and we softened the claim in the latest update.

---

### Official Review · AnonReviewer1 · 2018-10-30
**Limited novelty, nice write-up, impressive results, little/no analysis**

**Rating:** 5
**Confidence:** 5

**Review:**

My main reservation with this paper is the limited novelty. The approach seems to be a rather direct application of a subset of the sluice network architecture in [0] - which has been available on ArXiV since 2017 - with MUSE pre-trained embeddings. In particular, I don’t think the claim that the authors “propose the first zero-resource multilingual transfer learning model” is necessary - and I think it is way too strong a claim. Training an LSTM on English data with MUSE/vecmap embeddings is pretty standard by now, and this does not require any target language training data or cross-lingual supervision either. See zero-shot scenarios in [1-2], for example.

Apart from that, I think the write-up is nice, the approach makes a lot of sense, and results are impressive. I would have liked to see a bit more analysis. In particular, the fact that you learn gate values, makes it easy to analyze/visualize what and how your networks learn to share.

I think there’s a few baselines in between BWE and MAN, e.g., simple adversarial training and adversarial training with GradNorm [3], that would put your results in perspective. Finally, I would like to encourage the authors to run experiments with actual low-resource languages: A literature on cross-lingual transfer experimenting with German, Spanish, and Japanese, could end up being very heavily biased. For tasks with data in more languages, consider, for example, POS tagging [4], morphological analysis [5], or machine translation [6].

[0] https://arxiv.org/abs/1705.08142
[1] http://aclweb.org/anthology/P18-1074
[2] http://aclweb.org/anthology/P18-2063
[3] https://arxiv.org/abs/1711.02257
[4] http://universaldependencies.org/
[5] http://unimorph.org/
[6] http://christos-c.com/bible/

---

> ### Author Response · Authors · 2018-11-13
> **Response to the reviewer's comments**
>
> Thank you for your comments and feedback.
>
> Our method is similar to the Sluice Network (and other work on multi-task learning) in that we both try to learn what subspaces of features/parameter to share between tasks (in our case between languages) but it is also different in several fundamental ways.
> Most importantly, they use a global \alpha matrix to determine how to share information between various tasks statically on a task level, where two different samples from the same task would be treated in the same way. In contrast, our MoE model dynamically decide what to share on a token-level for any given input sample, where the model is able to dynamically use distinct expert mixtures for two samples from the same language.
> In addition, their method focuses on the multi-task learning setting, while ours deals with the multi-source transfer learning case.
> Finally, the shared features in our model is learned by language-adversarial training, which is absent in Sluice Network.
>
> We agree that it is not necessary to emphasize our method being the first zero-resource CLTL method, and we removed the claim in the latest update.
>
> We added the visualization of gate outputs to the paper in the latest update. One complication is that the expert gate makes decision at the token level instead of at the sentence level. That means the weights vary from token to token even in the same input sentence. It is hence harder to visualize. We did some aggregation to see if there were any insights on the language or sentence level.
>
> For other adversarial training baselines, we experimented with alternative adversarial training methods, such as the WGAN-WeightClip and the WGAN-GP training, but found the result to be similar with standard MAN. We did not further explore more possibilities for the MAN training part, but those can be readily adopted by our model if certain training technique is found helpful.
>
> We reached out to the authors of Xie et al. 2018 for their NER data on the low-resource language, Uyghur, but they were unable to share the data due to license issue. We will explore other possibilities of adding more experiments on low-resource languages.

---

### Official Review · AnonReviewer3 · 2018-11-02
**Interesting idea!**

**Rating:** 6
**Confidence:** 4

**Review:**

This paper presents a multilingual NLP model which performs very well on a target language with any leveraging labeled data. The authors evaluated their framework on there different tasks: slot filling, named entity recognition and text classification. Overall, the results look very promising.
- Strengthens:
+ The proposed idea is novel.
+ The results are very good for all three tasks.
- Weaknesses:
+ The authors claimed that their model knows what to share. However, they did not provide any evidence proving this hypothesis. Only the experimental results are not enough.
+ The paper also lacks an analysis to show to some extent what the model learned, e.g. the attention weights or the value of the gate. Is there any correlation between the similarity among languages (source and target) and the attention weights.
- What are not clear:
+ It is not clear to me what exactly has been done with the CharCNN embeddings in Section 4.2? How did the authors train the embeddings (only with the source languages or also with the target language)? It seems to me that the proposed model did not work well in this case.

---

> ### Author Response · Authors · 2018-11-13
> **Response to the reviewer's comments**
>
> Thank you for your comments and feedback.
>
> Regarding the weaknesses:
> - We showed detailed ablation analysis (Section 4.1.2) that closely matched our hypothesis that the MAN-MoE model learns what features to share across different languages dynamically. In particular, when transferring to the less similar language Chinese, the model with MAN removed performs significantly worse, indicating language-invariant features are important for this case. On the other hand, when transferring to German or Spanish, where the target language is more similar to a subset of source languages, we observe that the model with MoE removed performs much worse, illustrating the importance of private features in such cases. The fact that MAN-MoE outperforms both MAN-only and MoE-only in both cases show that our model is able to learn what's important for sharing in every scenario.
>
> - We added the visualization of gate outputs to the paper in the latest update. One complication is that the expert gate makes decision at the token level instead of at the sentence level. That means the weights vary from token to token even in the same input sentence. It is hence harder to visualize. We did some aggregation to see if there were any insights on the language or sentence level.
>
> For clarification:
> The CharCNN produces a character-level word embedding for each word, and is concatenated to the pretrained word embeddings. The CharCNN is randomly initialized and is updated end to end together with the rest of the model, and is not separately trained. Therefore, it is trained on the target language during the MAN training, since the adversarial training part uses monolingual data from all languages including the target language.
> We clarified this in the paper.

---

### Public Comment · ~Jiateng_Xie1 · 2018-11-06
**A missing citation**

Dear authors, we would like to point out that a citation to our recent paper at EMNLP '18 that does cross-lingual NER under the same setting as Section 4.2 and achieves slightly better results (at the time of this comment) is missing. Link: https://arxiv.org/abs/1808.09861
Thank you!

---

> ### Author Response · Authors · 2018-11-13
> **Response to your comments**
>
> Thank you for bringing this paper to our attention, and we added the citation and comparison in the latest update.
> Our results are similar with the ones obtained in this paper when it uses similar unsupervised embeddings (BWET (adv.) + self-att. in Table 1), despite that we did not use the CRF decoding layer.
> Furthermore, our method is a general multilingual model transfer approach and we experimented on multiple tasks in addition to NER.
> Please excuse of negligence of this paper since this paper was made available on arXiv only a few weeks before the submission deadline while the official EMNLP proceedings actually came after it.

---

### Comment · Area_Chair1 · 2018-11-19
**Reviewers: Please consider author response**

Dear Reviewers,

The authors have posted a response, and I'd appreciate if you could take a look at it and see if it has addressed any of your concerns. Thank you!

---

### Meta-Review · Area_Chair1 · 2018-12-13
**Nice results, but somewhat limited novelty and framing compared to previous work**

**Confidence:** 4
**Recommendation:** Reject

**Metareview:**

The proposed method proposes a new architecture that uses mixture of experts to determine what to share between multiple languages for transfer learning. The results are quite good.

There is still a bit of a lack of framing compared to the large amount of previous work in the field, even after initial revisions to cover reviewer comments. I think that probably this requires a significant rewrite of the intro and maybe even title of the paper to scope the contributions, and also make sure in the empirical analysis that the novel contributions are evaluated independently on their own (within the same experimental setting and hyperparameters).

As such, and given the high quality bar of ICLR, I can't recommend this paper be accepted at this time, but I encourage the authors to revise this explanation and re-submit a new version elsewhere.